# Selective dendritic localization of mRNA in *Drosophila* mushroom body output neurons

Jessica Mitchell[1], Carlas S Smith[1,2], Josh Titlow[3], Nils Otto[1], Pieter van Velde[2], Martin Booth[1,4], Ilan Davis[3], Scott Waddell[1]*

[1]Centre for Neural Circuits and Behaviour, University of Oxford, Oxford, United Kingdom; [2]Delft Center for Systems and Control, Delft University of Technology, Delft, Netherlands; [3]Department of Biochemistry, University of Oxford, Oxford, United Kingdom; [4]Department of Engineering Science, University of Oxford, Oxford, United Kingdom

**Abstract** Memory-relevant neuronal plasticity is believed to require local translation of new proteins at synapses. Understanding this process requires the visualization of the relevant mRNAs within these neuronal compartments. Here, we used single-molecule fluorescence in situ hybridization to localize mRNAs at subcellular resolution in the adult *Drosophila* brain. mRNAs for subunits of nicotinic acetylcholine receptors and kinases could be detected within the dendrites of co-labeled mushroom body output neurons (MBONs) and their relative abundance showed cell specificity. Moreover, aversive olfactory learning produced a transient increase in the level of *CaMKII* mRNA within the dendritic compartments of the γ5β'2a MBONs. Localization of specific mRNAs in MBONs before and after learning represents a critical step towards deciphering the role of dendritic translation in the neuronal plasticity underlying behavioral change in *Drosophila*.

*For correspondence:
scott.waddell@cncb.ox.ac.uk

Competing interests: The authors declare that no competing interests exist.

## Introduction

Memories are believed to be encoded as changes in the efficacy of specific synaptic connections. Dendritic localization of mRNA facilitates specificity of synaptic plasticity by enabling postsynaptic synthesis of new proteins where and when they are required (*Holt et al., 2019*). Visualizing individual dendritically localized mRNAs in memory-relevant neurons is therefore crucial to understanding this process of neuronal plasticity.

Single-molecule fluorescence in situ hybridization (smFISH) enables cellular mRNAs to be imaged at single-molecule resolution through the hybridization of a set of complementary oligonucleotide probes, each labeled with a fluorescent dye. Recent improvements in smFISH permit mRNA transcripts to be visualized in the dense heterogenous tissue of intact *Drosophila* brains (*Long et al., 2017*; *Yang et al., 2017*). Combining whole fly brain smFISH with neuron-specific co-labeling makes *Drosophila* an ideal model to investigate cell-specific mRNA localization and whether it is regulated in response to experience.

Olfactory learning in *Drosophila* depresses cholinergic synaptic connections between odor-specific mushroom body Kenyon cells (KCs) and mushroom body output neurons (MBONs) (*Barnstedt et al., 2016*; *Cohn et al., 2015*; *Handler et al., 2019*; *Hige et al., 2015*; *Owald et al., 2015*; *Perisse et al., 2016*; *Séjourné et al., 2011*). This plasticity is driven by dopaminergic neurons whose presynaptic terminals innervate anatomically discrete compartments of the mushroom body, where they overlap with the dendrites of particular MBONs (*Aso et al., 2010*; *Aso et al., 2014*; *Burke et al., 2012*; *Claridge-Chang et al., 2009*; *Li et al., 2020*; *Lin et al., 2014*; *Liu et al., 2012*). Dopamine-driven plasticity is mediated by cAMP-dependent signaling and associated kinases such

as calcium/calmodulin-dependent protein kinase II (CaMKII) and protein kinase A (PKA) (*Boto et al., 2014*; *Handler et al., 2019*; *Hige et al., 2015*; *Kim et al., 2007*; *Qin et al., 2012*; *Tomchik and Davis, 2009*; *Yu et al., 2006*; *Zhang and Roman, 2013*). Here, we demonstrate localization of mRNAs in the 3D volumes of MBON dendrites by registering smFISH signals with co-labeled neurons using a custom image analysis pipeline. Moreover, we find that aversive learning transiently elevates dendritic *CaMKII* transcript levels within γ5β'2a MBONs.

## Results and discussion

### mRNA localization in the intact adult *Drosophila* brain

Mammalian CaMKII mRNA is transported to neuronal dendrites, where it is locally translated in response to neuronal activity (*Bagni et al., 2000*; *Miller et al., 2002*; *Ouyang et al., 1999*). *Drosophila* CAMKII is critical for behavioral plasticity (*Griffith, 1997*; *Malik et al., 2013*) and is also thought to be locally translated (*Ashraf et al., 2006*). However, fly *CAMKII* mRNAs have not been directly visualized within individual neurons. We therefore first hybridized *CaMKII* smFISH probes to whole-mount brains and imaged the mushroom body (MB) calyx (*Figure 1A, B*), a recognizable neuropil containing the densely packed dendrites of ~2000 KCs and their presynaptic inputs from ~350 cholinergic olfactory projection neurons (*Bates et al., 2020a*) using a standard spinning disk confocal microscope. To detect and quantify mRNA within the 3D volume of the brain, we developed a FIJI-compatible custom-built image analysis tool that segments smFISH image data and identifies spots within the 3D volume using a probability-based hypothesis test. This enabled detection of mRNAs with a false discovery rate of 0.05. CaMKII smFISH probes labeled 56 ± 5 discrete puncta within each calyx (*Figure 1B, C*). In comparison, smFISH probes directed to the α1 nicotinic acetylcholine receptor (nAChR) subunit labeled 33 ± 2 puncta in the calyx (*Figure 1B, C*). Puncta were diffraction limited and the signal intensity distribution was unimodal (*Figure 1D–D'*), indicating that they represent single mRNA molecules.

### mRNA localization within MBON dendrites

*Drosophila* learning is considered to be implemented as plasticity of cholinergic KC-MBON synapses. To visualize and quantify mRNA specifically within the dendritic field of the γ5β'2a and γ1pedc>α/β MBONs, we expressed a membrane-tethered UAS-myr::SNAP reporter transgene using MBON-specific GAL4 drivers. This permitted simultaneous fluorescent labeling of mRNA with smFISH probes and the MBON using the SNAP Tag (*Figure 1E*). To correct for chromatic misalignment (*Matsuda et al., 2018*) that results from imaging heterogenous tissue at depth, we also co-stained brains with the dsDNA-binding dye Vybrant DyeCycle Violet (VDV). VDV dye has a broad emission spectrum so labeled nuclei can be imaged in both the SNAP MBON and smFISH mRNA channels. This triple-labeling approach allowed quantification and correction of any spatial mismatch between MBON and smFISH channels in x, y, and z planes, which ensures that smFISH puncta are accurately assigned within the 3D volume of the MBON dendritic field (*Figure 1F*).

Using this smFISH approach, we detected an average of 32 ± 2 *CaMKII* mRNAs (*Figure 1G, G'*) within the dendrites of γ5β'2a MBONs. However, in contrast to the calyx, we did not detect *nAChRα1* in γ5β'2a MBON dendrites (*Figure 1H, H'*). This differential localization of the *CaMKII* and *nAChRα1* mRNAs within neurons of the mushroom body is indicative of cell specificity. To probe mRNA localization in MBONs more broadly, we used a single YFP smFISH probe set and a collection of fly strains harboring YFP insertions in endogenous genes (*Lowe et al., 2014*). We selected YFP insertions in the *CaMKII*, *PKA-R2*, and *Ten-m* genes as test cases and compared the localization of their YFP-tagged mRNAs between γ5β'2a MBON and γ1pedc>α/β MBON dendrites.

The *CaMKII::YFP* allele is heterozygous in flies also expressing myr::SNAP in MBONs. Therefore, YFP smFISH probes detected half the number of *CaMKII* mRNAs in γ5β'2a MBON dendrites compared to *CaMKII*-specific probes (*Figure 2A, A', C*). Importantly, YFP probes hybridized to YFP-negative control brains produced background signal (*Figure 2B, B'*) that was statistically distinguishable in brightness from genuine smFISH puncta (*Figure 2D*). Comparing data from YFP-negative and YFP-positive samples allowed us to define the false discovery rate to be 14% when using YFP-directed probes (*Figure 2D*, *Figure 2—figure supplement 1*). These results indicate that the YFP probes are specific and that the YFP insertion does not impede localization of *CaMKII* mRNA. We

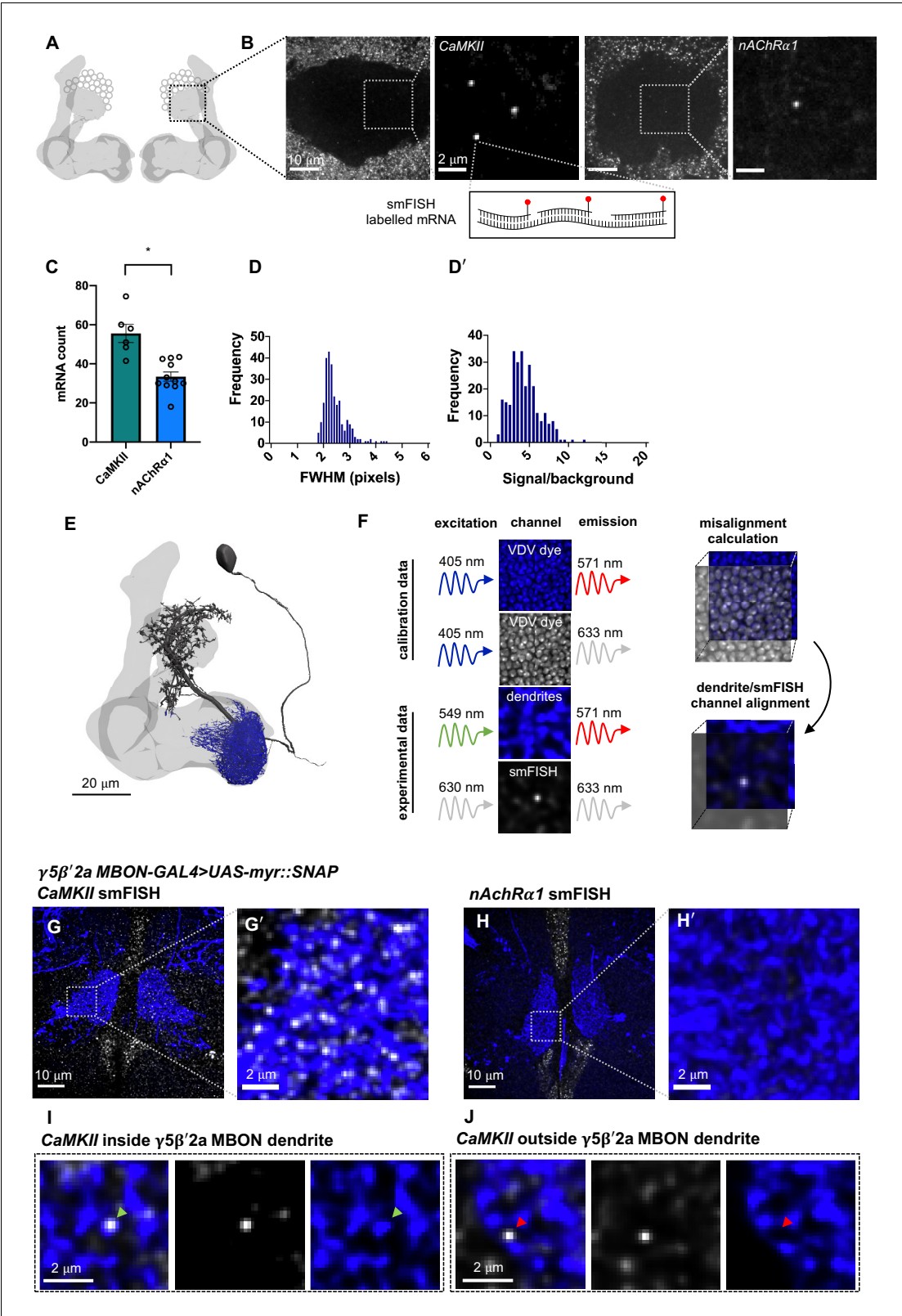

**Figure 1.** *CaMKII* and *nAChR* α1 mRNA visualized in the mushroom body (MB) calyx and γ5β'2a mushroom body output neuron (MBON) dendrites with single-molecule fluorescence in situ hybridization (smFISH). (**A**) Schematic of *Drosophila* MB. smFISH signal was imaged in the calyx, indicated by the dashed box. (**B**) *CaMKII* and *nAChRα1* mRNAs labeled with smFISH in the MB calyx. Images are maximum intensity projections of ten 0.2 μm z-sections. (**C**) More *CaMKII* mRNAs are detected in the MB calyx relative to *nAChRα1* (unpaired *t*-test: p=0.0003, t = 4.727, df = 15). (**D**) smFISH spot size

*Figure 1 continued on next page*

*Figure 1 continued*

distribution (full width half maximum, bottom) in MB calyx. (D'). Unimodal smFISH spot intensity distribution (signal/background) indicates imaging at single-molecule resolution. (E) Reconstruction of a γ5β'2a MBON (black) showing the dendritic field (blue) and MB (light gray). The projection to the contralateral MB is truncated. (F) Alignment of dendrite and smFISH imaging channels using co-labeling with dsDNA Vybrant DyeCycle Violet (VDV) dye. VDV is excited with 405 nm and emission is collected in the dendritic and smFISH imaging channels, which were then aligned in x, y, and z planes. (G, G') *CaMKII* smFISH within the γ5β'2a MBON dendrite co-labeled with R66C08-GAL4-driven UAS-myr::SNAP and visualized with JF547SNAP dye. Images are maximum intensity projections of ten 0.2 μm z-sections. (H, H') *nAchRα1* smFISH in γ5β'2aMBONs. Images are maximum intensity projections of ten 0.2 μm z-sections. (I) Single *CaMKII* smFISH puncta localized within a γ5β'2a MBON dendrite (green arrowhead). Images are single z-sections of 0.2 μm. (J) Single *CaMKII* smFISH puncta localized outside of the γ5β'2a MBON dendrite (red arrowhead). Images are single z-sections of 0.2 μm.

detected a similar abundance of *CaMKII::YFP* in the dendritic field of γ5β'2a (*Figure 2E*) and the γ1 dendritic region of γ1pedc>α/β (*Figure 2F*) MBONs (*Figure 2G*). In contrast, more *PKA-R2* mRNAs were detected in the dendrites of γ5β'2a MBONs compared to γ1pedc>α/β MBONs (*Figure 2G*). Importantly, the relative abundance of dendritically localized *CaMKII* and *PKA-R2* mRNAs did not simply reflect the levels of these transcripts detected in the MBON somata (*Figure 2H*). In addition, we did not detect *Ten-m* mRNAs in either γ5β'2a or γ1pedc>α/β MBON dendrites (*Figure 2G, I*), although they were visible in neighboring neuropil and at low levels in the MBON somata (*Figure 2H*). These results suggest that *CaMKII* and *PKA-R2* mRNAs are selectively localized to MBON dendrites.

Although we did not detect *nAChRα1* mRNA within γ5β'2a MBON dendrites, prior work has shown that nAChR subunits, including nAChRα1, are required in γ5β'2a MBON postsynapses to register odor-evoked responses and direct odor-driven behaviors (*Barnstedt et al., 2016*). Since the YFP insertion collection does not include nAChR subunits, we designed *nAChRα5* and *nAChRα6*-specific smFISH probes. These probes detected *nAchRα5* and *nAchRα6* mRNAs within γ5β'2a and γ1pedc>α/β MBON dendrites, with *nAchRα6* being most abundant (*Figure 2G*). Importantly, we detected *nAchRα1, nAchRα5,* and *nAchRα6* at roughly equivalent levels in the γ5β'2a and γ1pedc>α/β MBON somata (*Figure 2H*). Therefore, the selective localization of *nAchRα5* and *nAchRα6α6* mRNA to MBON dendrites indicates that these receptor subunits may be locally translated to modify the subunit composition of postsynaptic nAChR receptors.

Localized mRNAs were on average 2.8× more abundant in γ5β'2a relative to the γ1 region of γ1pedc>α/β MBON dendrites (*Figure 2G*). We therefore tested whether this apparent differential localization correlated with dendritic volume and/or the number of postsynapses between these MBONs. Using the recently published electron microscope volume of the *Drosophila* 'hemibrain' (*Scheffer et al., 2020*; *Figure 2E, F*), we calculated the dendritic volume of the γ5β'2a MBON to be 1515.36 nm$^3$ and the γ1 region of the γ1pedc>α/β MBON to be 614.20 nm$^3$. In addition, the γ5β'2a regions of the γ5β'2a MBON dendrite contain 30,625 postsynapses, whereas there are only 17,020 postsynapses in the γ1 region of the γ1pedc>α/β MBON. Larger dendritic field volume and synapse number is therefore correlated with an increased number of localized *nAchRα5, nAchRα6,* and *PKA-R2* mRNAs. The correlation, however, does not hold for *CaMKII* mRNA abundance. Selective localization of mRNAs to MBON dendrites therefore appears to be more nuanced than simply reflecting the size of the dendritic arbor, the number of synapses, or the level of transcripts detected throughout the cell.

## Learning transiently changes *CAMKII* mRNA abundance in γ5β'2a MBON dendrites

We tested whether *CaMKII::YFP* mRNA abundance in γ5β'2a and γ1pedc>α/β MBONs was altered following aversive learning (*Figure 3A, B*). We also quantified mRNA in the somata and nuclei of these MBONs (*Figure 3A, B'*). Transcriptional activity is indicated by a bright nuclear transcription focus (*Figure 3C*, *Figure 3—figure supplement 1*). We initially subjected flies to four conditions (*Figure 3D*): (1) an 'untrained' group that was loaded and removed from the T-maze but not exposed to odors or shock; (2) an 'odor only' group, exposed to the two odors as in training but without shock; (3) a 'shock only' group that was handled as in training and received the shock delivery but no odor exposure; and (4) a 'trained' group that was aversively conditioned by pairing one

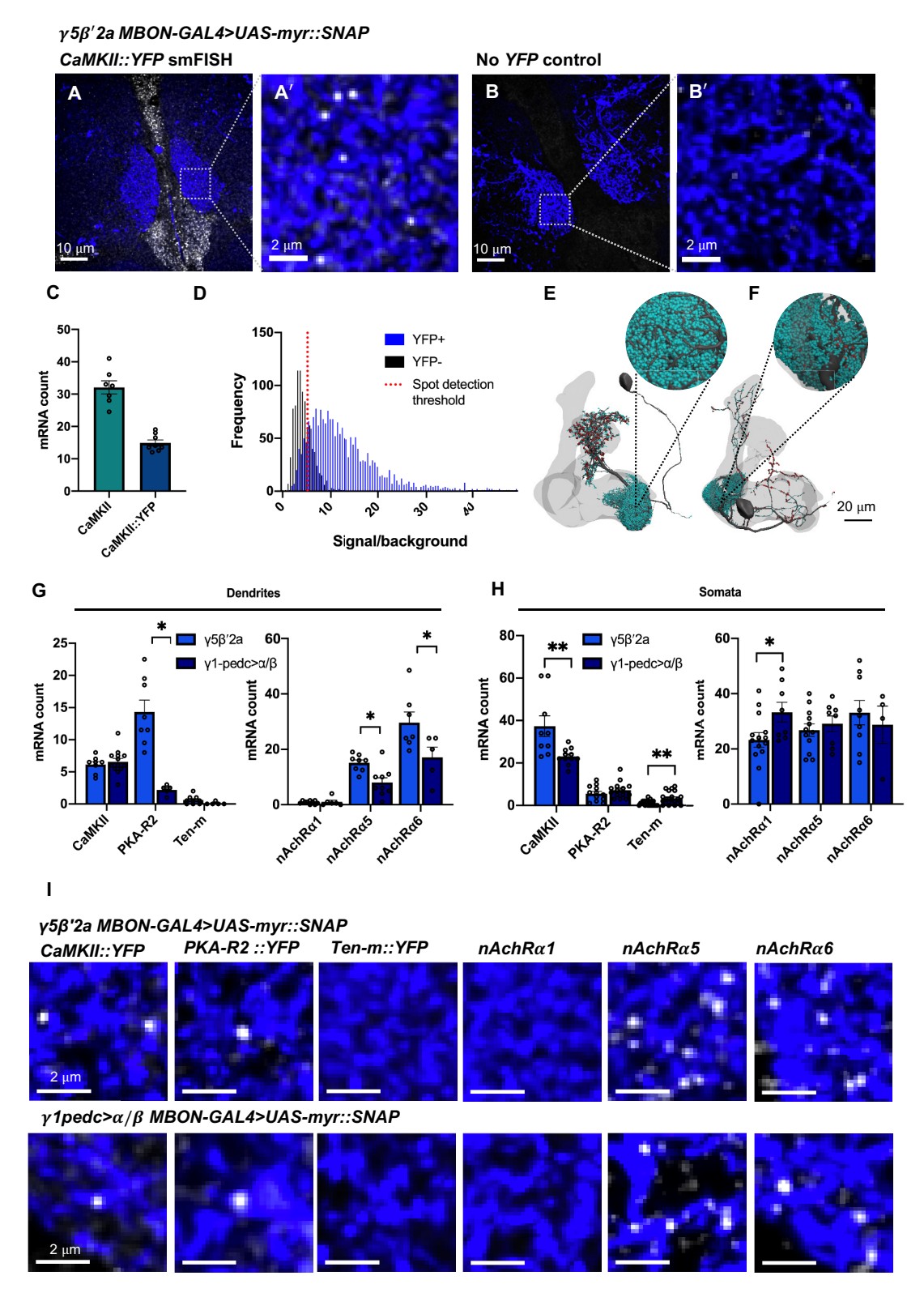

**Figure 2.** Differential localization of mRNAs in γ5β'2a and γ1pedc>α/β mushroom body output neuron (MBON) dendrites. (**A, A'**). *CaMKII::YFP* mRNA visualized in γ5β'2a MBON dendrites using YFP single-molecule fluorescence in situ hybridization (smFISH) probes. The γ5β'2a MBON is labeled by R66C08-GAL4-driven UAS-myr::SNAP and visualized with JF547SNAP dye. Images are maximum intensity projections of ten 0.2 μm z-sections. (**B, B'**). YFP smFISH signal in a γ5β'2a MBON in a negative control fly. Images are maximum intensity projections of ten 0.2 μm z-sections. (**C**) The *CaMKII::YFP*

*Figure 2 continued on next page*

*Figure 2 continued*

allele is heterozygous, resulting in detection of half as many *CaMKII* mRNAs in γ5β′2a MBONs using YFP probes relative to that detected with *CaMKII* gene-specific probes. (D) Signal/background intensity distribution of YFP probe signals in *CaMKII::YFP* brains relative to control brains with no threshold on signal detection. The signal/background intensity threshold for quantitative analyses (dotted red line) resulted in a false discovery rate of ≤14% (indicated by the overlap of the histograms on the right side of the dotted red line) (see also *Figure 2—figure supplement 1*). (E) Reconstruction of a γ5β′2a MBON. Individual postsynapses (turquoise spheres) and presynapses (red spheres) are labeled. The projection to the contralateral mushroom body (MB) is truncated. (F) Reconstruction of a γ1pedc>α/β MBON. Individual postsynapses (turquoise spheres) and presynapses (red spheres) are labeled. The projection to the contralateral MB is truncated. (G) Quantification of mRNA localization in γ5β′2a and γ1pedc>α/β MBON dendrites with YFP smFISH probes and gene-specific nicotinic acetylcholine receptor (nAChR) subunit smFISH probes. More *PKA-R2* transcripts localize within the dendrites of γ5β′2a MBONs relative to γ1pedc>α/β MBONs (unpaired *t*-test: p=0.004, t = 5.069, df = 11). *Ten-m* mRNAs did not localize to either MBON dendritic field. *CaMKII* mRNAs were detected in equal abundance. *nAChRα1* mRNAs did not localize to the dendrites of either γ5β′2a or γ1pedc>α/β MBONs. More *nAChRα5* (unpaired *t*-test: p=0.004, t = 3.368, df = 15) and *nAChRα6* (unpaired *t*-test: p=0.046, t = 2.274, df = 10) mRNAs localized to γ5β′2a MBON dendrites relative to γ1pedc>α/β MBON dendrites. (H) Quantification of mRNA in γ5β′2a and γ1pedc>α/β MBON somata with YFP smFISH probes and gene-specific nAChR subunit smFISH probes. More *CaMKII* transcripts were present within γ5β′2a MBON somata relative to γ1pedc>α/β MBON somata (unpaired *t*-test: p=0.0061, t = 3.103, df = 18). More *Ten-m* (Mann–Whitney test: p=0.0093, Mann–Whitney *U* = 120) and *nAChRα1* (unpaired *t*-test: p=0.0359, t = 2.250, df = 20) transcripts were detected in γ1pedc>α/β MBON somata relative to γ5β′2a MBON somata. (I) Example smFISH images of mRNAs localized in γ5β′2a (R66C08-GAL4>UAS-myr::SNAP) and γ1pedc>α/β MBON (MB112C-GAL4>UAS-myr::SNAP) dendrites. Images are maximum intensity projections of ten 0.2 μm z-sections. Asterisks denote significant difference (p<0.05). Data are means ± standard error of mean. Individual data points are displayed.

The online version of this article includes the following figure supplement(s) for figure 2:

**Figure supplement 1.** Effect of spot detection threshold on false-positive detections.

of the two odors with shock. Fly brains were extracted 10 min, 1 hr, or 2 hr after training and processed for smFISH.

*CaMKII* mRNA increased significantly in γ5β′2a MBON dendrites 10 min after training (*Figure 3E*) compared to all control groups. Including an additional 'unpaired' experiment, where odor and shock presentation was staggered, confirmed that the increase at 10 min after training requires coincident pairing of odor and shock (*Figure 3—figure supplement 2*). Moreover, levels returned to baseline by 1 hr and remained at that level 2 hr after training (*Figure 3E*). *CaMKII* mRNAs in γ5β′2a MBON somata showed a different temporal dynamic, with transcripts peaking 1 hr after training, albeit only relative to untrained and odor only controls (*Figure 3E*). The proportion of γ5β′2a nuclei containing a *CaMKII* transcription focus did not differ between treatments (*Figure 3E*), suggesting that the transcript increase in the somata is not correlated with the number of actively transcribing γ5β′2a nuclei, at least at the timepoints measured. In addition, the mean brightness of γ5β′2a transcription foci did not change across treatments (*Figure 3G*), although the variation was substantial. An increase of dendritically localized *CaMKII* mRNAs could result from enhanced trafficking or through the release of transcripts from protein bound states, which would increase smFISH probe accessibility and hence spot brightness (*Buxbaum et al., 2014*). Since the brightness of *CaMKII* mRNA spots detected in the dendrites of γ5β′2a MBONs did not change with treatment (*Figure 3H*), we conclude that the increased abundance likely results from altered traffic.

Assessing *CaMKII* mRNA abundance in γ1pedc>α/β MBONs after learning did not reveal a change in mRNA abundance in the dendrites or somata between trained flies and all control groups at all timepoints measured (*Figure 3F*). These results indicate specificity to the response observed in the γ5β′2a MBONs.

Since CaMKII protein is also labeled with YFP in *CaMKII::YFP* flies, we assessed protein expression by measuring YFP fluorescence intensity specifically within the MBON dendrites. This analysis did not reveal a significant difference in fluorescence intensity across treatments (*Figure 3—figure supplement 2*). However, since smFISH provides single-molecule estimates of mRNA abundance, a similar level of single-molecule sensitivity may be required to detect subcellular resolution changes in protein copy number. Moreover, new synthesis and replacement of specific isoforms of CaMKII could radically change local kinase activity (*Kuklin et al., 2017*; *Zalcman et al., 2018*), even without an observable change in overall abundance.

Early studies in *Drosophila* demonstrated that broad disruption of CAMKII function impaired courtship learning (*Broughton et al., 2003*; *Griffith et al., 1994*; *Griffith et al., 1993*; *Joiner and Griffith, 1997*). In contrast, later studies that manipulated activity more specifically in olfactory projection neurons or particular classes of KCs reported a preferential loss of middle-term or long-term

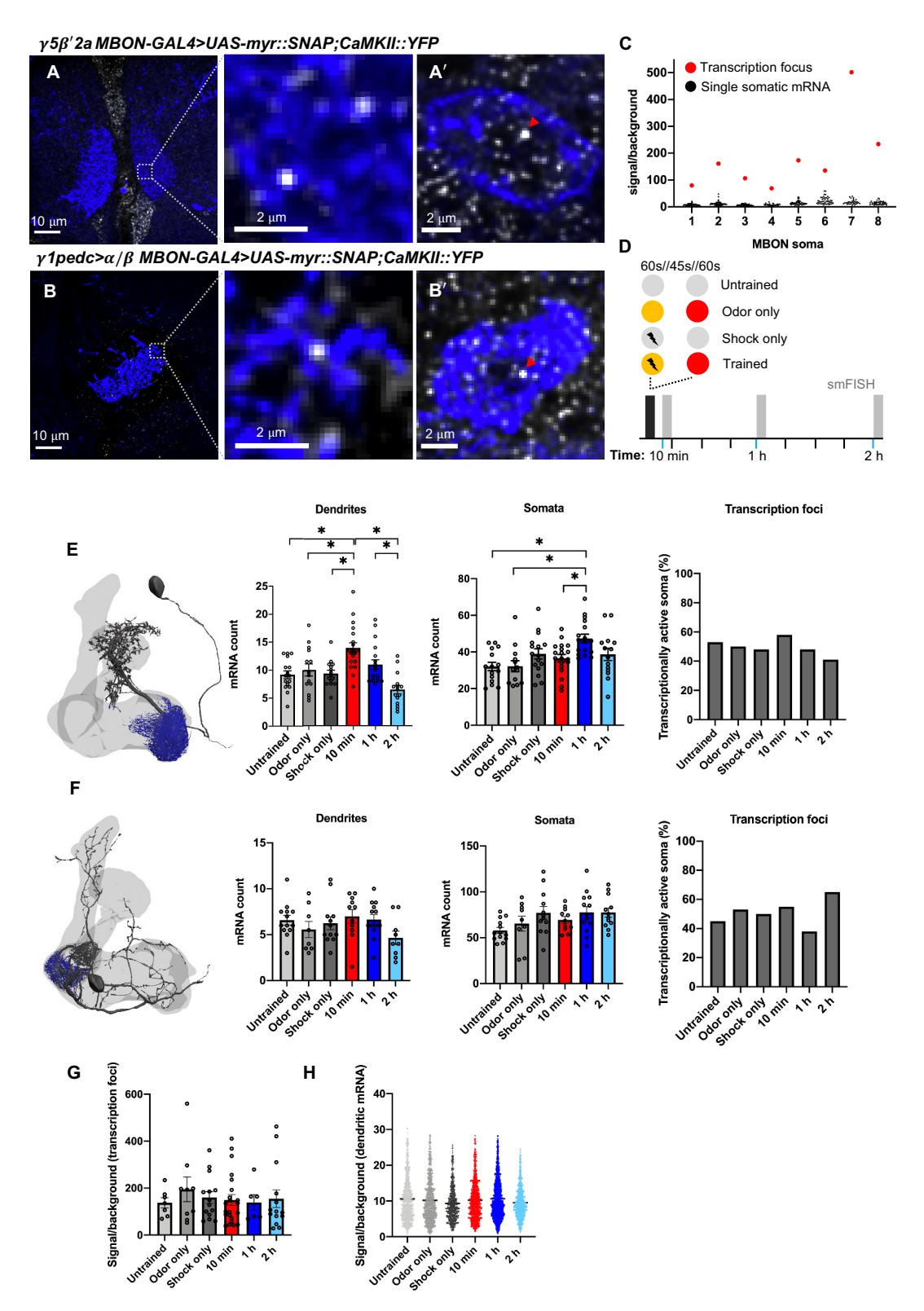

**Figure 3.** Learning alters *CaMKII* mRNA abundance in the γ5β'2a mushroom body output neurons (MBONs). (**A, A'**). *CaMKII::YFP* single-molecule fluorescence in situ hybridization (smFISH) in γ5β'2a MBON dendrites and soma (R66C08-GAL4>UAS-myr::SNAP). Images are maximum intensity projections of ten 0.2 μm z-sections. (**B, B'**). *CaMKII::YFP* smFISH in γ1pedc>α/β MBON dendrites and soma (MB112C-GAL4>UAS-myr::SNAP). Nuclear transcription foci are indicated (red arrowheads). Images are maximum intensity projections of ten 0.2 μm z-sections. (**C**) *CaMKII::YFP* smFISH signal/

*Figure 3 continued on next page*

*Figure 3 continued*

background in transcriptionally active γ5β'2a somata. Transcription foci are readily distinguished as the brightest puncta in the soma/nucleus (red data points). Note that only one transcription focus can be visualized per cell since the *CaMKII::YFP* allele is heterozygous. (D) Schematic of aversive training and control protocols followed by smFISH. The yellow and red circles represent the two odors. (E) *CaMKII::YFP* mRNA numbers in γ5β'2a MBON dendrites increase 10 min after odor–shock pairing, relative to control groups (one-way ANOVA: untrained-10 min p=0.001; odor only-10 min p=0.016; shock only-10 min p=0.002), and decrease to baseline by 2 hr (one-way ANOVA: 10 min-2 h p<0.001; 1–2 h p=0.004). *CaMKII::YFP* mRNA numbers in γ5β'2a MBON somata increase 1 hr after odor–shock pairing, relative to untrained (one-way ANOVA: p=0.001), odor only (one-way ANOVA: p=0.002), and 10 min post training (one-way ANOVA: p=0.025). The proportion of transcriptionally active γ5β'2a MBON somata is unchanged ($X^2$=2.064, df = 5, p=0.840). (F) *CaMKII::YFP* mRNA numbers are not changed by aversive odor–shock pairing in γ1pedc>α/β MBON dendrites (one-way ANOVA: f = 1.473, p=0.212), their somata (one-way ANOVA: f = 2.183, p=0.067), and there is no detected change in *CaMKII::YFP* transcription ($X^2$=3.723, df = 5, p=0.59). (G) Signal/background ratio of *CaMKII::YFP* transcription foci in γ5β'2a MBON somata. (H) Signal/background ratio of *CaMKII::YFP* mRNA localized in γ5β'2a MBON dendrites. Asterisks denote significant difference (p<0.05). Data are means ± standard error of mean. Individual data points are displayed.

The online version of this article includes the following figure supplement(s) for figure 3:

**Figure supplement 1.** Single-molecule fluorescence in situ hybridization (smFISH) labels different numbers of active *CaMKII* loci in homozygous and heterozygous flies.

**Figure supplement 2.** Further unpaired control and quantification of CaMKII::YFP after learning.

olfactory memory (*Akalal et al., 2010*; *Ashraf et al., 2006*; *Malik et al., 2013*). Here, we focused our analyses on two subtypes of MBONs, which are known to exhibit changes in odor-evoked activity after a single trial of aversive olfactory conditioning. Whereas γ1pedc>α/β MBON responses to the previously shock-paired odor are depressed immediately after aversive learning (*Hige et al., 2015*; *Perisse et al., 2016*), prior studies observed a learning-related increase of the conditioned odor response of γ5β'2a MBONs (*Bouzaiane et al., 2015*; *Owald et al., 2015*), likely resulting from a release of feedforward inhibition from γ1pedc>α/β MBONs (*Felsenberg et al., 2018*; *Perisse et al., 2016*). We therefore speculate that the specific change in *CaMKII* mRNA abundance in the γ5β'2a MBONs after aversive learning might be a consequence of network-level potentiation of their activity, such as that that would result from a release from inhibition. Since CAMKII local translation-dependent plasticity is expected to underlie more extended forms of memory (*Giese and Mizuno, 2013*; *Miller et al., 2002*), it will be interesting to investigate whether the training-evoked change in *CaMKII* mRNA abundance in the γ5β'2a MBON dendrites contributes to later aversive memory formation and maintenance. This may be possible with MBON-specific targeting of CAMKII mRNAs that contain the long 3'UTR, which is essential for dendritic localization and activity-dependent local translation (*Aakalu et al., 2001*; *Kuklin et al., 2017*; *Mayford et al., 1996*; *Rook et al., 2000*).

# Materials and methods

## Key resources table

| Reagent type (species) or resource | Designation | Source or reference | Identifiers | Additional information |
|---|---|---|---|---|
| Gene (*Drosophila melanogaster*) | *CaMKII* | NCBI | Gene ID: 43828 | |
| Gene (*Drosophila melanogaster*) | *PKA-R2* | NCBI | Gene ID: 36041 | |
| Gene (*Drosophila melanogaster*) | *Ten-m* | NCBI | Gene ID: 40464 | |
| Gene (*Drosophila melanogaster*) | *nAChRα1* | NCBI | Gene ID: 42918 | |
| Gene (*Drosophila melanogaster*) | *nAChRα5* | NCBI | Gene ID: 34826 | |
| Gene (*Drosophila melanogaster*) | *nAChRα6* | NCBI | Gene ID: 34304 | |

*Continued on next page*

*Continued*

| Reagent type (species) or resource | Designation | Source or reference | Identifiers | Additional information |
|---|---|---|---|---|
| Genetic reagent (*Drosophila melanogaster*) | R66C08-GAL4 | Bloomington *Drosophila* Stock Center (**Owald et al., 2015**) | RRID:BDSC_49412 | |
| Genetic reagent (*Drosophila melanogaster*) | MB112c-GAL4 | Bloomington *Drosophila* Stock Center (**Perisse et al., 2016**) | RRID:BDSC_68263 | |
| Genetic reagent (*Drosophila melanogaster*) | UAS-myr::SNAPf | Bloomington *Drosophila* Stock Center | RRID:BDSC_58376 | |
| Genetic reagent (*Drosophila melanogaster*) | CaMKII::YFP | Kyoto Stock Centre (**Lowe et al., 2014**) | RRID:DGGR_115127 | |
| Genetic reagent (*Drosophila melanogaster*) | PKA-R2::YFP | Kyoto Stock Centre (**Lowe et al., 2014**) | RRID:DGGR_115174 | |
| Genetic reagent (*Drosophila melanogaster*) | Ten-m::YFP | Kyoto Stock Centre (**Lowe et al., 2014**) | RRID:DGGR_115131 | |
| Chemical compound | 20% v/v paraformaldehyde | Thermo Fisher Scientific | Cat#15713S | |
| Chemical compound | RNase-free 10× PBS | Thermo Fisher Scientific | Cat#AM9625 | |
| Chemical compound | Triton X-100 | Sigma-Aldrich | Cat#T8787 | |
| Chemical compound | 20× RNase-free SSC | Thermo Fisher Scientific | Cat#AM9763 | |
| Chemical compound | Deionized formamide | Thermo Fisher Scientific | Cat#AM9342 | |
| Chemical compound | 50% dextran sulphate | Millipore | Cat#S4030 | |
| Chemical compound | Vybrant DyeCycle Violet Stain | Thermo Fisher Scientific | Cat#V35003 | |
| Chemical compound | Vectashield anti-fade mounting medium | Vector Laboratories | Cat#H-1000-10 | |
| Chemical compound | JF549-SNAPTag | **Grimm et al., 2015** | | |
| Chemical compound | Mineral oil | Sigma-Aldrich | Cat#M5904 | |
| Chemical compound | 4-Methocyclohexanol (98%) | Sigma-Aldrich | Cat#218405 | |
| Chemical compound | 3-Octanol (99%) | Sigma-Aldrich | Cat#153095 | |
| Software, algorithm | FIJI | NIH (**Schindelin et al., 2012**) | http://fiji.sc/ | |
| Software, algorithm | MATLAB R2019b | The MathWorks, Natick, MA | https://www.mathworks.com/products/matlab.html | |
| Software, algorithm | GraphPad Prism 8 | GraphPad Software, La Jolla, CA | https://www.graphpad.com/scientific-software/prism/ | |
| Software, algorithm | *Drosophila* brain smFISH analysis | This paper (**Mitchell, 2021**) | see Data availability section | |
| Software, algorithm | Blender | Blender Foundation, Amsterdam | https://www.blender.org | |
| Software, algorithm | NAVis 0.2.0 | **Bates et al., 2020b** | https://pypi.org/project/navis/ | |

## Fly strains

Flies were raised on standard cornmeal agar food in plastic vials at 25°C and 40–50% relative humidity on a 12 hr:12 hr light:dark cycle. Details of fly strains are listed in the Key Resources Table.

## smFISH probes

Oligonucleotide probe sets were designed using the web-based probe design software https://www.biosearchtech.com/stellaris-designer. The YFP smFISH probe set was purchased from LGC BioSearch Technologies (CA, USA) prelabeled with Quasar-670 dye. *CaMKII, nAChRα1, nAChRα5,* and *nAChRα6* DNA oligonucleotide sets were synthesized by Sigma-Aldrich (Merck) and enzymatically

labeled with ATTO-633 according to *Gaspar et al., 2017*. DNA oligonucleotide sequences for each smFISH probe set are provided in Supplementary file 1.

## Whole *Drosophila* brain smFISH

Whole adult brain smFISH was performed essentially as described (*Yang et al., 2017*). The 2–4-day-old adult *Drosophila* brains were dissected in 1× phosphate buffered saline (PBS) and fixed in 4% v/v paraformaldehyde for 20 min at room temperature. Brains were washed 2× with PBS, followed by 20 min in 0.3% v/v Triton X-100 in PBS (PBTX) to permeabilize the tissue, then 15 min in PBTX with 500 nM JF549-SNAPTag (*Grimm et al., 2015*) for neuronal labeling. Then, 3 × 10 min washes in PBTX removed excess dye. Samples were then incubated in wash buffer (2× RNase-free SSC + 10% v/v deionized formamide) for 10 min at 37°C, wash buffer was replaced with hybridization buffer (2× RNase-free SSC, 10% v/v deionized formamide, 5% w/v dextran sulphate, 250 nM smFISH probes), and samples incubated overnight at 37°C. Hybridization buffer was removed before samples were washed 2× in freshly prepared wash buffer and incubated 40 min in wash buffer containing Vybrant DyeCycle Violet Stain (1:1000) to label nuclei. Samples were then washed 3× times in wash buffer, mounted on a glass slide covered with Vectashield anti-fade mounting medium (refractive index 1.45), and immediately imaged.

## Olfactory conditioning

Aversive olfactory conditioning was performed essentially as described by *Tully and Quinn, 1985*. 3-Octanol (OCT) was used as the shock-paired odor. 4-Methylcyclohexanol (MCH) was used as the unpaired odor. Odors were prepared at concentrations of 9 µl OCT in 8 ml mineral oil, and 8 µl MCH in 8 ml mineral oil. Groups of ~100 flies were aliquoted into plastic vials containing standard cornmeal agar food and a 2 × 6 cm piece of filter paper. Flies were conditioned as follows: 1 min OCT paired with 12 × 90 V shocks at 5 s interstimulus interval; 45 s clean air; 1 min MCH. Control groups were handled in the same way except for the differing presentation of either odors or shock. Untrained flies experienced no odor or shock, the odor only group experienced the two odor presentations without shock, and the shock only group received the shock presentations but no odors. Aversive olfactory conditioning was performed at 23 °C and 70% relative humidity. Following training, flies were returned to food vials and brains were dissected either 10 min, 1 hr, or 2 hr later, and smFISH analyses performed.

For the 'unpaired' experiment, the interval between presentations was extended from 45 to 180 s to avoid trace conditioning of the unpaired odor. In the trained group, flies were presented with 1 min OCT paired with 12 × 90 V shocks at 5 s interstimulus interval, 180 s clean air, and then 1 min MCH. In the unpaired group, flies received 12 × 90 V shocks at 5 s interstimulus interval (no odor pairing), 180 s clean air, and then 1 min MCH. Other control groups were handled in the same way except that the odor only group experienced the two odor presentations without shock and the shock only group received the shock presentations but no odors. Following training, flies were returned to food vials and brains were dissected 10 min later for smFISH analyses.

## Microscopy

Samples were imaged on a spinning disk confocal microscope (Perkin Elmer UltraView VoX) with a 60× 1.35 N.A. oil immersion UPlanSApo objective (Olympus) and a filter set to image fluorophores in DAPI, FITC, TRITC, and CY5 channels (center/bandwidth; excitation: 390/18, 488/24, 542/27, 632/22 nm; emission: 435/48, 594/45, 676/34 nm), the corresponding laser lines (488/4.26, 561/6.60, 640/3.2, 405/1.05, 440/2.5, 514/0.8 nm/mW), and an EMCCD camera (ImagEM, Hamamatsu Photonics). The camera pixel size is 8.34 µm, resulting in a pixel size in image space of approximately 139 nm. Optical sections were acquired with 200 nm spacing along the z-axis within a 512 × 512 pixel (71.2 × 71.2 µm) field of view.

## Deconvolution

Deconvolution was carried out using commercially available software (Huygens Professional v19.10.0p1, SVI Delft, The Netherlands). Raw image data generated in .mvd2 file format were converted to OME.tiff format using FIJI (*Schindelin et al., 2012*) (convert_mvd2_to_tif.ijm). Spherical aberration was estimated from the microscope parameters (see Microscopy). Z-dependent

momentum preserving deconvolution (CLME algorithm, theoretical high-NA PSF, iteration optimized with quality change threshold 0.1% and iterations 40 maximum, signal-to-noise ratio 20, area radius of background estimation is 700 nm, a brick mode is 1 PSF per brick, single array detector with reduction mode SuperXY) was then applied to compensate for the depth-dependent distortion in point spread function, thereby reducing artifacts and increasing image sharpness.

### Multi-channel alignment

Misalignment between channels was corrected for using Chromagnon (v. 0.81) (*Matsuda et al., 2018*). To estimate channel misalignment, nuclei were labeled with the broad emission spectrum dye (Vybrant DyeCycle Violet Stain, Thermo Fisher) (*Smith et al., 2015*). The dye was excited at 405 nm, and emission was recorded using the appropriate filters for each imaging channel. Chromatic shift was estimated by finding the affine transformation that delivers a minimum mean square error between the nuclear stain in the various channels. Nuclear calibration channels for chromatic shift correction were separated using ImageJ (see macro Split_ometiff_channels_for_chromcorrect.ijm). The affine transformation was estimated and alignment was performed by calling Chromagnon from Python (see script chromagnon_bash.py). The resulting aligned and deconvolved images were saved in .dv format for further downstream analysis.

### Calculating postsynaptic abundance and volume of γ5β'2a and γ1pedc>α/β MBON dendrites

Neuromorphological calculations were performed with NAVis 0.2.0 library functions in Python (https://pypi.org/project/navis/) (*Bates et al., 2020a*) using data obtained from the *Drosophila* hemibrain dataset (v.1.1) (https://neuprint.janelia.org) (*Scheffer et al., 2020*). To calculate the dendritic volume and postsynaptic abundance of γ5β'2a and γ1pedc>α/β MBONs, neuron skeletons, neuropil meshes, and synapse data were first imported. Neural skeletons were then used to generate 3D neuron reconstructions. Dendritic processes of the γ5β'2a MBON were determined by intersecting neuronal skeletons with the MB mesh containing the γ5 and β'2a compartments. Dendritic processes of the γ1pedc>α/β MBON were determined by intersecting the skeleton within the γ1 MB compartment mesh. The available γ1 MB compartment mesh did not encompass the entirety of the γ1pedc>α/β MBON dendrites in the γ1 MB compartment, so the volume of the mesh was scaled up 1.35×. This intersects with almost all γ1pedc>α/β MBON dendrites in the γ1 MB compartment, but not any other substantial part of the neuron. Dendritic volume ($nm^3$) was calculated as the sum of the neurite voxels multiplied by $8^3$ since the resolution of each voxel is 8 $nm^3$. The number of postsynapses within these compartments was also determined using the synapse data that accompany the neuron skeletons (*Scheffer et al., 2020*).

Data visualization smFISH data were visualized in FIJI (*Schindelin et al., 2012*). Maximum intensity projections representing 2 μm sections are presented for visualization purposes. *Figure 1I and J* are single z-sections (representing a 0.2 μm section). The 3D reconstructions of γ5β'2a and γ1pedc>α/β MBONs were created in Blender v.2.8.2 with NAVis 0.2.0 plug-in and using data obtained from http://www.neuprint.janelia.org.

### mRNA detection

An smFISH spot detection MATLAB script based on *Smith et al., 2015* was written to quantify localized mRNA transcripts in *Drosophila* brains. Software for processing smFISH datasets is available as Supplementary Software. The smFISH channel was extracted and stored as a 3D grayscale image. mRNA signal was detected using 3D generalized likelihood ratio test (*Smith et al., 2015*). The false detection rate is 0.05, and the spot width is $\sigma_{x,y}$ = 1.39 and $\sigma_z$ = 3.48. After 3D detection, the intensity, background, width, and subpixel position of the detected mRNA spots are estimated using maximum likelihood estimation (MLE) (*Smith et al., 2010*).To reduce the impact of overlapping spots in 3D, only a 2D cross section is used from the z-plane where the spot is detected. To filter out spurious detections, all spots with a width >5 pixels are discarded.

### mRNA-dendrite co-localization

To quantify calyx and dendritic localized smFISH puncta, the calyx and dendritic area were first segmented manually. The contour of the calyx and dendritic area is converted to a mask ($M_1$) using the

MATLAB R2019b function roipoly. To quantify smFISH puncta co-localizing with dendrite label, a mask of the dendrite label is created by enhancing the image using a difference of Gaussians filter (width of 1 and 5 pixels) and then thresholding the product between the enhanced image ($A$) and masked area ($M_1$) to obtain a mask ($M_2$):

$$M_2 = A \circ M_1 > \text{mean}(A \circ M_1) + \text{std}(A \circ M_1)$$

where mean() and std() are the sample mean and sample standard deviation of the image intensity values, and $A \circ B$ is the Hadamard product between A and B. The sample standard deviation is calculated as

$$std(x) = \sqrt{\frac{1}{N-1}\sum_{i=1}^{N}(x_i - \bar{x})^2}$$

where $N$ is the number of data points. smFISH signal within γ5β'2a MBON dendrites innervating the γ5 and β'2a MB compartments was analyzed. smFISH signal within the γ1pedc>α/β dendrites innervating the γ1 MB compartment was analyzed. Sections of $10 \times 0.2$ μm individual z-slices of MB calyx, γ5β'2a MBON dendrites, or γ1pedc>α/β MBON dendrites were analyzed. smFISH puncta overlapping with the calyx or dendrite mask were considered co-localizing and therefore localized within that neuronal compartment.

## Spot brightness and full width half maximum (FWHM) analysis

For each detection, a region of interest (ROI) is extracted as a 2D box in the x–y plane with a size of $2\times(3\sigma_{x,y}+1)$. For each ROI, the MLE of the x and y position, the number of photons, the number of background photons, and the width of the 2D Gaussian, $\sigma_{x,y}$, is computed. The FWHM of the spots is calculated as $FWHM = 2\sqrt{2ln(2)}\ \sigma_{x,y}$.

## Verification of transcription foci

Soma containing bright nuclear transcription foci were selected to quantify the difference in intensity relative to diffraction-limited smFISH puncta. The nuclear localization of the smFISH puncta with the highest photon count was validated by visual inspection and considered to correspond to the transcription site. The width ($\sigma_{x,y}$) of the transcription foci significantly differs from the sparse smFISH signal and is estimated by fitting a 2D Gaussian to the transcription site using the MATLAB 2019b nonlinear least-squares routine *lsqcurvefit*. Transcription foci brightness and background were computed using the same MLE protocol as for diffraction-limited spots, but with the estimated $\sigma_{x,y}$.

## YFP fluorescence intensity

To quantify YFP fluorescence intensity within co-labeled neurons, we developed a FIJI-compatible macro plug-in. Depth-dependent bleaching was first corrected for over the z-stack using an exponential fit. Background signal was then subtracted in each z-section using a rolling ball filter with a width of 60 pixels. Five z-sections above and below the center of the image were cropped for analysis. YFP fluorescence intensity was recorded within the dendrites or soma of the co-labeled neuron using the mask described above (mRNA-dendrite co-localization). Fluorescence intensity was calculated as analog digital units (adu)/volume (dendrites or soma) to give adu/voxel. Software for analyzing fluorescent protein expression in single neurons is available as Supplementary Software.

## Statistical analyses

Data were visualized and analyzed statistically using GraphPad Prism version 8.3.1 (332). The distribution of a dataset was assessed with a Shapiro–Wilk test. Gaussian distributed smFISH abundance was compared between two groups using an unpaired *t*-test. Gaussian distributed smFISH abundance between multiple groups was compared using a one-way ANOVA followed by Tukey's post hoc test. Non-Gaussian distributed smFISH abundance was compared between two groups using a Mann–Whitney $U$ test. Proportions of transcriptionally active soma were compared to transcriptionally inactive soma using a chi-square test. YFP-positive and -negative smFISH intensity distributions were compared with a two-sided Wilcoxon rank-sum test. YFP fluorescence intensity across

treatments was compared using a one-way ANOVA for Gaussian distributed data and a Kruskal–Wallis test for non-Gaussian distributed data. Statistical significance is defined as $p < 0.05$.

## Acknowledgements

We are grateful for the microscopy facilities and expertise provided by Micron Advanced Bioimaging Unit (supported by Wellcome Strategic Awards 091911 and 107457). We thank Jeff Lee for assistance with smFISH probe generation and members of the Waddell group for discussion. JM was funded through the BBSRC Interdisciplinary Bioscience Doctoral Training Programme. CSS and PVV were funded by the Netherlands Organisation for Scientific Research (NWO), under NWO START-UP project no. 740.018.015 and NWO Veni project no. 16761. CSS was initially supported by grants to SW and MB and acknowledges a research fellowship through Merton College, Oxford, UK. MB was funded by Wellcome Strategic Awards (095927 and 107457) and the MRC/EPSRC/BBSRC (MR/K01577X/1). ID was supported by a Wellcome Senior Research Fellowship (096144), Wellcome Trust Investigator Award (209412), and Wellcome Strategic Awards (091911 and 107457). SW was funded by a Wellcome Principal Research Fellowship (200846/Z/16/Z), an ERC Advanced Grant (789274), and a Wellcome Collaborative Award (203261/Z/16/Z).

## Additional information

### Funding

| Funder | Grant reference number | Author |
| --- | --- | --- |
| Wellcome Trust | 200846/Z/16/Z | Scott Waddell |
| Wellcome Trust | 203261/Z/16/Z | Scott Waddell |
| European Research Council | | Scott Waddell |
| Netherlands Organisation for Scientific Research | NWO START-UPproject no. 740.018.015 | Carlas S Smith Pieter van Velde |
| Wellcome Trust | 107457 | Ilan Davis Martin Booth |
| Wellcome Trust | 096144 | Ilan Davis |
| Wellcome Trust | 209412 | Ilan Davis |
| Wellcome Trust | 091911 | Ilan Davis |
| BBSRC | | Jessica Mitchell |
| NWO | Veni project no. 16761 | Carlas S Smith Pieter van Velde |
| MRC/EPSRC/BBSRC | MR/K01577X/1 | Martin Booth |

The funders had no role in study design, data collection and interpretation, or the decision to submit the work for publication.

### Author contributions

Jessica Mitchell, Conceptualization, Data curation, Formal analysis, Validation, Investigation, Visualization, Methodology, Writing - original draft, Writing - review and editing; Carlas S Smith, Conceptualization, Resources, Data curation, Software, Formal analysis, Supervision, Validation, Investigation, Visualization, Methodology, Project administration, Writing - review and editing; Josh Titlow, Formal analysis, Investigation, Methodology, Writing - review and editing; Nils Otto, Formal analysis, Investigation, Visualization, Methodology, Writing - review and editing; Pieter van Velde, Software, Formal analysis, Investigation; Martin Booth, Supervision, Funding acquisition; Ilan Davis, Resources, Funding acquisition, Writing - review and editing; Scott Waddell, Conceptualization, Resources, Supervision, Funding acquisition, Writing - original draft, Project administration, Writing - review and editing

## Author ORCIDs

Nils Otto (iD) http://orcid.org/0000-0001-9713-4088
Pieter van Velde (iD) http://orcid.org/0000-0002-7281-8026
Ilan Davis (iD) http://orcid.org/0000-0002-5385-3053
Scott Waddell (iD) https://orcid.org/0000-0003-4503-6229

## Decision letter and Author response

Decision letter https://doi.org/10.7554/eLife.62770.sa1
Author response https://doi.org/10.7554/eLife.62770.sa2

# Additional files

## Supplementary files

• Supplementary file 1. Oligonucleotide sequences of single-molecule fluorescence in situ hybridization (smFISH) probe sets.

• Transparent reporting form

## Data availability

Pipeline code and the User Manual are available in the GitHub repository at https://github.com/qnano/smFISHlearning copy archived at https://archive.softwareheritage.org/swh:1:rev:c73d1b977c767256982b40736f42b87d940caf05/. An example dataset of raw and processed images is available at https://figshare.com/articles/dataset/Example_data/13568438. All other processed and raw datasets that support the findings of this study are available at https://doi.org/10.6084/m9.figshare.13573475.

The following datasets were generated:

| Author(s) | Year | Dataset title | Dataset URL | Database and Identifier |
|---|---|---|---|---|
| Mitchell J, Smith C, Titlow J, Otto N, Velde P, Booth MJ, Davis I, Waddell S | 2021 | Data for figures of Selective dendritic localization of mRNA in *Drosophila* Mushroom Body Output Neurons | https://doi.org/10.6084/m9.figshare.13573475 | figshare, 10.6084/m9.figshare.13573475 |
| Velde P | 2021 | Example data for Dendritic localization of mRNA in *Drosophila* Mushroom Body Output Neurons | https://figshare.com/articles/dataset/Example_data/13568438 | figshare, 10.6084/m9.figshare.13568438 |

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
