## [Decision Letter]

**Acceptance summary:**

This work describes the implementation of smFISH to visualise mRNAs in memory-encoding, mushroom-body output neurons (MBONs) in vivo. Using this technique, the authors report that levels of a *CaMKII* reporter mRNA are increased after associative training in dendrites of these MBONs known to mediate this form of memory. This is unexpected because gene expression changes are thought to occur after treatments that induce long-term memory and the increase is seen to occur after a single training trial, which should not induce LTM. Thus, the work points to unexpected and previously undocumented changes in the state or levels of dendritic mRNA after training regimens that do not induced long-term memory. Together, this work describes a very useful technology to study memory-associated changes in RNA transport, localisation, or accessibility in vivo and points to unexpected regulation of these processes by experiences that do not cause LTM. Additional analyses to establish the function and mechanisms for this learning-dependent *CamKII* localization localization will be required to establish the importance of these intriguing observations. The current work provides a foundation for such future studies.

**Decision letter after peer review:**

[Editors’ note: the authors submitted for reconsideration following the decision after peer review. What follows is the decision letter after the first round of review.]

Thank you for submitting your work entitled "Dendritic localization of mRNA in *Drosophila* Mushroom Body Output Neurons" for consideration by *eLife*. Your article has been reviewed by three peer reviewers, and the evaluation has been overseen by a Reviewing Editor and a Senior Editor. The reviewers have opted to remain anonymous.

Our decision has been reached after consultation between the reviewers. Based on these discussions and the individual reviews below, we genuinely regret to inform you that this potentially important work will not be considered further for publication in *eLife*.

All of the reviewers agreed that this was a technically impressive paper working towards an exciting and important goal. However there was a clear consensus not only that the observations made required more controls to firm up, but also and more critically, need to be extended further, to establish the origin of the increased smFISH signal (how), as well as its biological significance (why).

Reviewer #1:

Targeting of mRNAs to synapses, combined with activity-dependent local translation, has been proposed to underlie various forms of synaptic plasticity, as well as formation of long-term memories. How this process is regulated in vivo during physiological learning and memory has remained unclear.

Here, the authors aimed at addressing this question by studying dendritic mRNA localization in *Drosophila* brains. This is relevant, as imaging, conditioning and manipulation of neuronal activity can be efficiently combined in this system, yet no quantitative analysis of dendritic RNA localization has so far been reported. In this study, the authors describe that distinct mRNAs localize to different extent to the dendrites of neurons undergoing learning and memory-dependent plasticity (γ5β'2a MBON and γ1pedc>α/β MBON). Furthermore, they uncover that aversive olfactory conditioning induces within minutes a transient increase in the amount of *camkII* mRNA molecules localizing to the dendrites of γ5β'2a MBON neurons. This last observation is interesting, as it suggests the existence of plasticity-dependent regulatory mechanisms controlling dendritic mRNA localization. However, both the origin of such a regulatory process and its biological implications remain unclear (see major points below) and should be investigated further.

1) In Figure 2, the authors compare the dendritic localization profile of different RNAs in different neuron types (γ5β'2a and γ1pedc>α/β MBONs).

– In 2F, they compare the accumulation of RNAs encoding subunits of the nAchR receptor. This is interesting as it points to local and specific regulation of receptor composition.

– In 2E, the rationale to compare the distribution of *CamkII*, *PKA-R2* and *Ten-m* mRNAs, however, is less clear: why are *PKA-R2* and *Ten-m* specifically analyzed/interesting? Why is it surprising to not see *Ten-m* RNAs in the dendrites of γ5β'2a MBON? Also, it is difficult (impossible?) to interpret the observed differences in dendritic localization without knowing 1- if the transcripts are at all expressed in the neurons under consideration, and 2- their expression levels. Addressing this last point could be done by counting the number of transcripts found in the corresponding cell bodies (as done in Figure 3).

– Last, the authors compare the amount of RNAs in γ1pedc>α/β and γ5β'2a MBONs and try to make the point i- that the amount of dendritically-localized mRNAs correlate with dendritic volume and synapse number, and ii- that these factors may thus be "important determinants of localized mRNA copy number". Clearly, however, 2 of the 4 RNAs they identify as localized do not follow this principle (*CamkII* is equally abundant in both neuron types and the fold difference observed for *PKA-R2* is not in the range of the observed difference in dendritic volume/synapse number). This makes the correlation quite weak. Determining whether the amount of RNAs present in dendrites correlate with the total amount of RNA (or at least the amount in cell bodies) for each species in the two populations under consideration may make a stronger case and highlight trends and/or specific behaviors.

2) The increase in the amount of dendritically-localized Camk2 RNA seen 10 minutes after conditioning is the most interesting observation of this study. This observation should however be consolidated as both the biological meaning of the transient increase, and the mechanisms underlying this regulatory process remain unclear.

– Investigating further if the observed changes are linked to local translation (and not only changes in global protein levels), or are specific to short-term/long-term memory paradigms would significantly strengthen the manuscript. This may also allow the authors to explain how their data fit with previous experiments demonstrating that the training protocol used in this study induces translation-independent short-term memory.

– Alternatively, understanding how this process is regulated (activity-dependent? 3'UTR-dependent? transport vs degradation…) may help the authors point to interesting underlying mechanisms.

Reviewer #2:

Mitchell et al. applied smFISH on the mushroom body circuit to visualize seemingly single- mRNA signals localized to the dendrites of single MBONs. They further found that aversive olfactory learning increases *CamKII* mRNA levels in the dendrites. The topic is interesting, and the imaging technique is high-level. But there are concerns regarding the evidence to support their claims.

1) For example in Figure 2D, there is a considerable amount of false detection of background signal in YFP- condition. The authors should estimate the false detection rate. Importantly, the intensity of the signals and the background would be different with different probes. It would be necessary to verify the signal by using mutants or genetic knockdown for each probe.

2) To claim selective localization of mRNA (Figure 2F), it is not enough to count the mRNA puncta in the dendrites because that may simply reflect the transcription level. Therefore, it is necessary to count mRNA in reference regions (e.g. soma as they did in Figure 3) and quantify the relative localization.

3) In Figure 3E and F, unpaired training control is necessary to claim that the increase is learning-dependent. Also, it is not fair to compare the 10 min-odor only to 1 hr-trained (Figure 3E). The authors should compare the trained and the unpaired at the respective time points.

Reviewer #3:

This technically accomplished and well-written manuscript documents dendritic localization of subsets of mRNAs in specific neurons within the *Drosophila* brain and its modulation by learning paradigms. The development of a robust system for studying mRNA localization in a tractable physiological setting is valuable. However, the biological insights provided by this study fall short of what I would expect for a short communication in *eLife*. Specifically, the authors are unable to find a correlation between learning-associated changes in mRNA distribution and the synthesis or distribution of the protein product. In the absence of such data, or other functional data, the relevance of mRNA localization in this system is unclear.

The authors provide evidence that changes in mRNA distribution are not accompanied by changes in the proportion of cells transcribing the mRNA. This suggests that localization rather than synthesis of the mRNA is primarily being affected, which is a key distinction. Given the importance of this part of the work the authors need to provide additional evidence that the large foci are indeed sites of transcription. For example, endogenous autosomal genes might be expected to have two foci, whereas X-linked genes should have one focus in males and two in females. The authors should also quantify the fluorescence intensity of presumptive nascent transcript foci with and without learning as it seems conceivable that changes in mRNA distribution are due to the same subset of cells transcribing more of the mRNA (rather than an increase in the proportion of transcriptionally active cells).

[Editors’ note: further revisions were suggested prior to acceptance, as described below.]

Thank you for submitting your article "Selective dendritic localization of mRNA in *Drosophila* mushroom body output neurons" for consideration by *eLife*. Your article has been reviewed by two peer reviewers, and the evaluation has been overseen by Mani Ramaswami as Reviewing Editor and K VijayRaghavan as the Senior Editor. The reviewers have opted to remain anonymous.

Essential Revisions:

1) One thing still unclear is the calculation of false detection rate (previous point #1). The authors claim that the rate is 5%, but how was this calculated? In the corresponding data (Figure 2D), “contamination” counts look much larger than 5%. If the authors mean the q-value of by the 5% of FDR, what was the multiple hypothesis testing? How is the value of "signal/background" in Figure 2D calculated? Justification is critical to interpret data of fewer counts (e.g. dendritic *CaMKII*). This needs to be clarified.

2) The Discussion should include a clear consideration of the implications and observed increase in reporter RNA levels in MBON dendrites. Why is it observed after training trials that should not induce LTM and how does this observation impact or fit with previous observations and conclusions? Are there any specific experiments that could test the authors models? In addition, a discussion of potential mechanisms that could underlie this phenomenon and experiments required to address them would be valuable to the reader.

---

## [Author Response]

[Editors’ note: The authors appealed the original decision. What follows is the authors’ response to the first round of review.]

Reviewer #1:Targeting of mRNAs to synapses, combined with activity-dependent local translation, has been proposed to underlie various forms of synaptic plasticity, as well as formation of long-term memories. How this process is regulated in vivo during physiological learning and memory has remained unclear.Here, the authors aimed at addressing this question by studying dendritic mRNA localization in *Drosophila* brains. This is relevant, as imaging, conditioning and manipulation of neuronal activity can be efficiently combined in this system, yet no quantitative analysis of dendritic RNA localization has so far been reported. In this study, the authors describe that distinct mRNAs localize to different extent to the dendrites of neurons undergoing learning and memory-dependent plasticity (γ5β'2a MBON and γ1pedc>α/β MBON). Furthermore, they uncover that aversive olfactory conditioning induces within minutes a transient increase in the amount of camkII mRNA molecules localizing to the dendrites of γ5β'2a MBON neurons. This last observation is interesting, as it suggests the existence of plasticity-dependent regulatory mechanisms controlling dendritic mRNA localization. However, both the origin of such a regulatory process and its biological implications remain unclear (see major points below) and should be investigated further.1) In Figure 2, the authors compare the dendritic localization profile of different RNAs in different neuron types (γ5β'2a and γ1pedc>α/β MBONs).– In 2F, they compare the accumulation of RNAs encoding subunits of the nAchR receptor. This is interesting as it points to local and specific regulation of receptor composition.– In 2E, the rationale to compare the distribution of CamkII, PKA-R2 and Ten-m mRNAs, however, is less clear: why are PKA-R2 and Ten-m specifically analyzed/interesting? Why is it surprising to not see Ten-m RNAs in the dendrites of γ5β'2a MBON? Also, it is difficult (impossible?) to interpret the observed differences in dendritic localization without knowing 1- if the transcripts are at all expressed in the neurons under consideration, and 2- their expression levels. Addressing this last point could be done by counting the number of transcripts found in the corresponding cell bodies (as done in Figure 3).

As previously stated in the manuscript, we studied CAMKII, *PKA-R2* and *Ten-m* mRNAs because we wished to determine the utility of using the YFP insertion library with the same YFP-targeted smFISH probes. PKA has of course been implicated in neuronal plasticity and *Ten-m* is another neuronally expressed gene, and therefore a potentially interesting thing to compare CAMKII and *PKA-R2* to. As requested by the reviewer, we now provide quantification of expression of these mRNAs in the somata of the relevant neurons.

– Last, the authors compare the amount of RNAs in γ1pedc>α/β and γ5β'2a MBONs and try to make the point i- that the amount of dendritically-localized mRNAs correlate with dendritic volume and synapse number, and ii- that these factors may thus be "important determinants of localized mRNA copy number". Clearly, however, 2 of the 4 RNAs they identify as localized do not follow this principle (CamkII is equally abundant in both neuron types and the fold difference observed for PKA-R2 is not in the range of the observed difference in dendritic volume/synapse number). This makes the correlation quite weak. Determining whether the amount of RNAs present in dendrites correlate with the total amount of RNA (or at least the amount in cell bodies) for each species in the two populations under consideration may make a stronger case and highlight trends and/or specific behaviors.

The reviewer raises an excellent point. We have now assessed the distribution of all the mRNAs we look at in both the somata and dendrites. We have now better described these results. Whereas the abundance of the *nAchR*α*5*, *nAchR*α*6* and *PKA-R2* mRNAs correlates to the size of the dendritic arbor and synpase number, the correlation does not hold for *CaMKII* mRNA abundance. We have therefore altered the conclusion accordingly.

2) The increase in the amount of dendritically-localized Camk2 RNA seen 10 minutes after conditioning is the most interesting observation of this study. This observation should however be consolidated as both the biological meaning of the transient increase, and the mechanisms underlying this regulatory process remain unclear.– Investigating further if the observed changes are linked to local translation (and not only changes in global protein levels), or are specific to short-term/long-term memory paradigms would significantly strengthen the manuscript. This may also allow the authors to explain how their data fit with previous experiments demonstrating that the training protocol used in this study induces translation-independent short-term memory.– Alternatively, understanding how this process is regulated (activity-dependent? 3'UTR-dependent? transport vs degradation…) may help the authors point to interesting underlying mechanisms.

We agree this is an interesting result that raises several questions. As we said in the manuscript there is currently no method available to assess local translation with a similar resolution to smFISH in the fly brain. Since this is currently impossible, we think it reasonable that it is beyond the scope of a Short Report. We already show that the increased signal is transient following a single training trial, going up at 10 min and down again by 1 h. Moreover, this does not happen in naïve (mock trained) flies, or those exposed to the odor or shock alone. In comparison signals in the somata go up transiently at 1h and are back down at 2h. Our data are therefore consistent with these changes occurring after learning. I think it’s potentially endless to request that we look after spaced training given that we see a transient increase following a single trial. At present this is a technically very demanding set of experiments conducted at multiple time points on flies that have been handled in four different ways. In the revision we have added an “unpaired control” which supports the notion that the increase in dendritic *CaMKII* mRNA is the result of learning.

Re: translation independent short-term memory. We are not suggesting that this CAMKII mRNA response is necessarily causal for memory. However, we show that it specifically occurs after training (as mentioned above we now show that it also does not occur in flies trained in an unpaired manner). As an aside, I am sure the reviewers and editors are aware that all experiments claiming protein synthesis independence of biological process are undermined by being unable to totally block it and maintain cellular and/or organismal viability.

Reviewer #2:Mitchell et al. applied smFISH on the mushroom body circuit to visualize seemingly single- mRNA signals localized to the dendrites of single MBONs. They further found that aversive olfactory learning increases CamKII mRNA levels in the dendrites. The topic is interesting, and the imaging technique is high-level. But there are concerns regarding the evidence to support their claims.1) For example in Figure 2D, there is a considerable amount of false detection of background signal in YFP- condition. The authors should estimate the false detection rate. Importantly, the intensity of the signals and the background would be different with different probes. It would be necessary to verify the signal by using mutants or genetic knockdown for each probe.

We have now determined the false detection rate for YFP smFISH probes to be <5%. We are aware that things could be different using different probes and that is one reason why this study aimed to utilise the same YFP probe set with the different flies carrying YFP insertions in interesting genes, eg. CAMKII, *PKA-R2* and Ten-m. Flies lacking YFP have been used as controls. Our control for other things we probed such as nAChR subunits is that their signals are differentially localised. For example, nAChRa1 was detected in somata but not in dendrites whereas nAChRa5 and nAChRa6 are in somata and differentially localized in dendrites. Mutants for nAChR receptors are unavailable/ lethal and RNAi would be hypomorphic, poorly controlled, and therefore likely unconvincing.

2) To claim selective localization of mRNA (Figure 2F), it is not enough to count the mRNA puncta in the dendrites because that may simply reflect the transcription level. Therefore, it is necessary to count mRNA in reference regions (e.g. soma as they did in Figure 3) and quantify the relative localization.

We agree and have now included data collected from the somata of the relevant neurons.

3) In Figure 3E and F, unpaired training control is necessary to claim that the increase is learning-dependent. Also, it is not fair to compare the 10 min-odor only to 1 hr-trained (Figure 3E). The authors should compare the trained and the unpaired at the respective time points.

We state that the increase occurs “after training”. We currently compare it to mock trained (untrained) flies and other flies exposed to odor alone, or shock alone. No changes are observed in these other flies. The only observed change we see is a transient increase of CAMKII that is specific to the γ5β'2a MBON after training. This suggests the increase results from pairing since it does not occur after either odor or shock exposure alone, that are presumably able to drive “activity”. Nevertheless, we have now performed another set of experiments (presented in Figure 3—figure supplement 2A) that include an “unpaired” control and show that the increase in dendritic *CaMKII* only occurs in trained flies.

Reviewer #3:This technically accomplished and well-written manuscript documents dendritic localization of subsets of mRNAs in specific neurons within the *Drosophila* brain and its modulation by learning paradigms. The development of a robust system for studying mRNA localization in a tractable physiological setting is valuable. However, the biological insights provided by this study fall short of what I would expect for a short communication in eLife. Specifically, the authors are unable to find a correlation between learning-associated changes in mRNA distribution and the synthesis or distribution of the protein product. In the absence of such data, or other functional data, the relevance of mRNA localization in this system is unclear.

We are not sure what “insights” one expects for a Short Report. We show dendritic localization of several mRNAs in memory-relevant locations in the fly brain for the first time. We also show that CAMKII localization is altered after training. It is true that the reason for the change in mRNA localization is currently unclear. However, “relevance” is a criticism that is easy to deliver and hard to contend with. Moreover, the fact that we did not observe significant changes in YFP fluorescence does not mean that there is no correlation between learning-associated changes in mRNA abundance and changes in protein abundance. Fluorescence intensity measurements for the protein are extremely crude, compared with the single molecule mRNA resolution that we have achieved using smFISH. The technology required to image translation events at molecular resolution in a whole brain does not yet exist.

The authors provide evidence that changes in mRNA distribution are not accompanied by changes in the proportion of cells transcribing the mRNA. This suggests that localization rather than synthesis of the mRNA is primarily being affected, which is a key distinction. Given the importance of this part of the work the authors need to provide additional evidence that the large foci are indeed sites of transcription. For example, endogenous autosomal genes might be expected to have two foci, whereas X-linked genes should have one focus in males and two in females. The authors should also quantify the fluorescence intensity of presumptive nascent transcript foci with and without learning as it seems conceivable that changes in mRNA distribution are due to the same subset of cells transcribing more of the mRNA (rather than an increase in the proportion of transcriptionally active cells).

These are good points. We have now provided information regarding fluorescence intensity of transcription foci, and show that we do not observe any changes. It is important to note that we already show that CAMKII-YFP flies have only one focus, which is expected as these flies are heterozygous for the allele. This is of course equivalent to the suggestion of X-linked, etc. We have also now included more data in Figure 3—figure supplement 1 that illustrates monoallelic and biallelic labelling of transcription foci.

[Editors’ note: what follows is the authors’ response to the second round of review.]

Essential Revisions:1) One thing still unclear is the calculation of false detection rate (previous point #1). The authors claim that the rate is 5%, but how was this calculated? In the corresponding data (Figure 2D), “contamination” counts look much larger than 5%. If the authors mean the q-value of by the 5% of FDR, what was the multiple hypothesis testing? How is the value of "signal/background" in Figure 2D calculated? Justification is critical to interpret data of fewer counts (e.g. dendritic CaMKII). This needs to be clarified.

We thank the reviewer for catching this important point. In the analysis of single molecule imaging data, there are two possible sources of false detection: (1) in the analytical detection of diffraction limited spots, and (2) from background signal of any unbound probes in the sample. The false detection rate (FDR) owing from spot detection (1) is assessed by performing a hypothesis test in each pixel i.e. is there a molecule present or not. If we would consider this as a single hypothesis testing problem with 5% false positive probability we would expect that 5% of all pixels are false positives. In order to assess the FDR owing from unbound probes (2), we ran the same detection algorithm on YFP positive, and negative control brains (2D). Since only spots with signal/background >6 are counted, the overlap between detections in the YFP positive and negative controls is 14%. As the reviewer notes, this is represented by the overlap to the right of the signal detection threshold line (red dotted line) of the YFP+ and YFP- histograms in Figure 2D. We have now included an additional supplementary figure (Figure 2—figure supplement 1) which shows how varying the signal detection threshold changes the overlap (i.e. the false detection rate) and also the number of spots which are included/excluded (i.e. counted/discarded). Increasing the spot detection threshold reduces the false detection rate, but also increases the number of detections that are discarded, including real spots. We chose the signal/background threshold >6 in an effort to balance the trade-off between throwing away real spots and falsely counting background noise. We believe the inclusion of this additional figure clarifies the reviewer comments regarding the false detection rate. The legend for Figure 2 and the main text have also been adjusted to clarify this issue.

2) The Discussion should include a clear consideration of the implications and observed increase in reporter RNA levels in MBON dendrites. Why is it observed after training trials that should not induce LTM and how does this observation impact or fit with previous observations and conclusions? Are there any specific experiments that could test the authors models? In addition, a discussion of potential mechanisms that could underlie this phenomenon and experiments required to address them would be valuable to the reader.

We have included the following paragraph in the Discussion:

“Early studies in *Drosophila* demonstrated that broad disruption of CAMKII function impaired courtship learning (Broughton et al., 2003; Griffith et al., 1994, 1993; Joiner and Griffith, 1997). […] This may be possible with MBON-specific targeting of CAMKII mRNAs that contain the long 3'UTR, which is essential for dendritic localization and activity-dependent local translation (Aakalu et al., 2001; Kuklin et al., 2017; Mayford et al., 1996; Rook et al., 2000).”